# Activation of Human Dendritic Cells by Ascophyllan Purified from *Ascophyllum nodosum*

**DOI:** 10.3390/md17010066

**Published:** 2019-01-19

**Authors:** Wei Zhang, Minseok Kwak, Hae-Bin Park, Takasi Okimura, Tatsuya Oda, Peter Chang-Whan Lee, Jun-O Jin

**Affiliations:** 1Scientific Research Center, Shanghai Public Health Clinical Center & Institutes of Biomedical Sciences, Shanghai Medical College, Fudan University, Shanghai 201508, China; weiwei061215@126.com; 2Department of Chemistry, Pukyong National University, Busan 48513, Korea; mkwak@pukyong.ac.kr; 3Marine-integrated Bionics Research Center, Pukyong National University, Busan 48513, Korea; 4Department of Medical Biotechnology, Yeungnam University, Gyeongsan 38541, Korea; ys02104@ynu.ac.kr; 5Research and Development Division, Hayashikane Sangyo Co., Ltd., Shimonoseki, Yamaguchi 750-8608, Japan; tokimura@hayashikane.co.jp; 6Graduate School of Science and Technology, Nagasaki University, Nagasaki 852-8521, Japan; t-oda@nagasaki-u.ac.jp; 7Department of Biomedical Sciences, University of Ulsan College of Medicine, ASAN Medical Center, Seoul 05505, Korea

**Keywords:** ascophyllan, dendritic cell activation, mitogen-activated protein kinase, T cell proliferation, peripheral blood dendritic cell

## Abstract

In our previous study, we showed that ascophyllan purified from *Ascophyllum nodosum* treatment promotes mouse dendritic cell (DC) activation in vivo, further induces an antigen-specific immune response and has anticancer effects in mice. However, the effect of ascophyllan has not been studied in human immune cells, specifically in terms of activation of human monocyte-derived DCs (MDDCs) and human peripheral blood DCs (PBDCs). We found that the treatment with ascophyllan induced morphological changes in MDDCs and upregulated co-stimulatory molecules and major histocompatibility complex class I (MHC I) and MHC II expression. In addition, pro-inflammatory cytokine levels in culture medium was also dramatically increased following ascophyllan treatment of MDDCs. Moreover, ascophyllan promoted phosphorylation of ERK, p38 and JNK signaling pathways, and inhibition of p38 almost completely suppressed the ascophyllan-induced activation of MDDCs. Finally, treatment with ascophyllan induced activation of BDCA1 and BDCA3 PBDCs. Thus, these data suggest that ascophyllan could be used as an immune stimulator in humans.

## 1. Introduction

In comparison to chemically synthesized molecules, natural products show low cytotoxicity in both animals and humans. Moreover, plant compounds such as sugars, microRNA and secondary metabolites have shown great bioactivity [1,2]. Many natural polysaccharides exhibit bioactivity such as anti-inflammatory, immune stimulatory, anti-viral, and anti-bacterial effects. Recent studies have demonstrated that marine natural polysaccharides such as fucoidan, carrageenan, and ascophyllan have an immune stimulatory capacity in mice [3,4,5]. Ascophyllan is purified from *Ascophyllum nodosum* and shows immune stimulatory effects in mice, especially the activation of spleen dendritic cell (DCs) and natural killer (NK) cells [4,6,7,8]. Moreover, a combined treatment of antigen and ascophyllan promotes antigen-specific immune responses, which further induces anti-cancer effects in mice in vivo [9]. Although the immune stimulatory effects of ascophyllan have well investigated in mice, the effects in human cells have not been studied.

When pathogens enter our body, immune cells are activated to protect the body [10]. Innate immune cells promote direct clearance of the pathogen and induce adaptive immune cell activation, such as T and B cells [10,11,12]. Macrophages and DCs are antigen presenting cells (APCs), which phagocytose pathogens and present antigens to T cells [10,12,13]. Compared to macrophages, which have limited function in the induction of antigen-specific immune activation due to their lack of antigen processing and presentation capacity, DCs are powerful APCs that control T cell proliferation and activation [14,15]. In a human DC study, peripheral blood monocytes were differentiated to MDDCs by treatment with granulocyte-macrophage colony-stimulating factor (GM-CSF) and interleukin-4 (IL-4), then cultured with immune stimulatory molecules [16,17]. These MDDCs showed different function and morphology compared to human PBDCs. Human PBDCs are comprised of two main subtypes: plasmacytoid and myeloid DCs. Following a viral infection, plasmacytoid DCs (pDCs) contribute to the production of type I interferons (IFNs) [18]. Myeloid DCs (mDCs) can be further divided into BDCA1 and BDCA3 PBDCs, where BDCA1 cells promote CD4 T cell activation and BDCA3 cells specialize in the induction of CD8 T cell stimulation [19,20,21]. Therefore, for the evaluation of novel immune stimulatory molecules, activation of human PBDC subsets is required.

Fucoidan is the most studied marine natural polysaccharide and has been shown to promote human PBDC activation [22]. Our previous study compared the immune stimulatory effects of ascophyllan and fucoidan in mouse DCs, but the effects of human DC activation have not been studied [4,9]. Ascophyllan can promote spleen DC activation in mice, and this effect is even stronger than that induced by fucoidan. We, therefore, hypothesize that ascophyllan may also be able to induce human DC activation and that this effect may be stronger than that of fucoidan. The present study was undertaken to test this hypothesis.

## 2. Results

### 2.1. Ascophyllan from Ascophyllum Nodosum Induced Activation of Monocyte-Derived Dendritic Cells (MDDCs)

Ascophyllan treatment has been shown to promote the maturation of spleen and lymph node DCs [4,9]. Here, we sought to evaluate the effect of ascophyllan on the activation of human monocyte-derived DCs (MDDCs). Moreover, the effect of ascophyllan was compared with that of fucoidan, the most studied immune stimulatory natural polysaccharide. The morphology of MDDCs was substantially changed following the treatment with ascophyllan (Figure 1A). In addition, CD80 and CD83 expression levels in MDDCs were dose-dependently increased by ascophyllan treatment, where doses of 50 μg/mL and 100 μg/mL showed similar effects (Figure 1B). Furthermore, expression levels of co-stimulatory molecules were significantly upregulated by ascophyllan compared to phosphate buffered saline (PBS) treatment (Figure 1C). The capacity of MDDC activation by ascophyllan was similar to those induced by fucoidan. These data show that ascophyllan can induce MDDC activation in vitro, at a dose of 50 μg/mL.

### 2.2. Activation of MDDCs by Ascophyllan Was Dependent on p38 Phosphorylation

To identify the signaling pathway responsible for MDDC activation by ascophyllan, we next examined the phosphorylation of mitogen-activated protein kinase (MAPK) following the treatment with ascophyllan. Ascophyllan treatment promoted phosphorylation of ERK, p38 and JNK (Figure 2A). Inhibitors of ERK, p38 and JNK were used to determine whether phosphorylation is required for ascophyllan-induced activation of MDDCs. We found that treatment with the p38 inhibitor, SB203580, suppressed the expression of co-stimulatory molecules that were induced by ascophyllan, while ERK (PD98059) and JNK inhibitors (SP600125) did not prevent the upregulation of these molecules in MDDCs (Figure 2B,C). These data suggest that the ascophyllan-induced activation of MDDCs is dependent on p38 phosphorylation.

### 2.3. Production of Pro-Inflammatory Cytokines by MDDCs Was Increased Following Ascophyllan Treatment

Activated DCs produce high levels of pro-inflammatory cytokines [23]. To evaluate the effect of ascophyllan on cytokine production by MDDCs, we treated MDDCs with 50 μg/mL ascophyllan for 24 h, and then measured the cytokine levels in the culture medium. Ascophyllan treatment induced an increase in interleukin-6 (IL-6), IL-12 and tumor necrosis factor-α (TNF-α) levels in the culture medium, and these levels were similar to those seen in the fucoidan-treated positive control (Figure 3A). Moreover, the ascophyllan-induced increase in cytokine production was almost completely inhibited by the p38 inhibitor, SB203580, but not by ERK and JNK inhibitors (Figure 3B). Thus, these data indicate that ascophyllan promotes pro-inflammatory cytokine production, and this is dependent on phosphorylation of p38.

### 2.4. Ascophyllan-Induced Activation of PBDC Subsets Was Dependent on p38 Signaling

Having shown that ascophyllan induces MDDC activation, we next examined whether ascophyllan can promote activation of PBDC subsets. As shown in Figure 4A, PBDC subsets were defined as BDCA1^+^ and BDCA3^+^ PBDCs in CD11c^+^lineage^−^ live leukocytes. Co-stimulatory molecules and MHC class I and II expression were measured in both BDCA1^+^ and BDCA3^+^ PBDCs after 24 h of ascophyllan treatment. Expression of co-stimulatory molecules and MHC class I and II was markedly increased by the ascophyllan treatment in both BDCA1^+^ and BDCA3^+^ PBDCs, which was similar to fucoidan-treated controls (Figure 4B). The upregulation of surface-active marker expression in PBDC subsets was inhibited by SB203580 treatment (Figure 4C). In addition, IL-6, IL-12 and TNF-α levels in peripheral blood mononuclear cells (PBMC) culture medium were also substantially upregulated following ascophyllan treatment (Figure 4D), which was abrogated by addition of SB203580 (Figure 4E). Thus, these data suggest that ascophyllan induces PBDC activation and pro-inflammatory cytokine production in PBMCs.

## 3. Discussion

Marine polysaccharides, including ascophyllan, fucoidan and carrageenan, have shown immune modulatory effects in humans and animals [3,4,5]. In a mouse study, administration of marine polysaccharides functions as immune suppressive molecules, especially oral administration of fucoidan and ascophyllan [8,24]. However, different administration routes, such as intraperitoneal and intravenous injection, show immune activation capacity by the polysaccharide [25,26,27]. In addition, fucoidan and ascophyllan promote spleen and lymph node (LN) DC activation and antigen-specific immune responses in mice in vivo, which further inhibits tumor growth [9,25]. In a human study, fucoidan has been shown to promote activation of MDDCs and PBDCs [22]. In the present study, we also demonstrated that ascophyllan can induce MDDC activation in vitro.

Previous studies showed that MAPK signaling contributes to the activation of DCs [28,29]. Lipopolysaccharide (LPS), a well-known immune stimulatory molecule, induces DC maturation, and this is dependent on MAPK signaling [28,29]. This function is mediated mainly by the upregulation of co-stimulatory molecules and pro-inflammatory cytokines in the DCs. ERK phosphorylation in MDDCs by LPS causes upregulation of pro-inflammatory cytokines [29]. Moreover, phosphorylation of JNK efficiently alters co-stimulatory molecule expression in LPS-stimulated MDDCs [30]. In LPS-induced MDDC activation, p38 phosphorylation is essential for both co-stimulatory molecule and pro-inflammatory cytokine production [31]. Treatment with ascophyllan also promoted phosphorylation of MAPK pathway components, including ERK, p38 and JNK. However, inhibition of ERK and JNK signaling did not suppress co-stimulatory molecule expression or pro-inflammatory cytokine production in ascophyllan-treated MDDCs. Phosphorylation of ERK and JNK was upregulated by ascophyllan, while inhibition of p38 almost completely suppressed pro-inflammatory cytokine production and co-stimulatory molecule expression in ascophyllan-stimulated MDDCs. The different effects of these signaling pathway inhibitors in ascophyllan- and LPS-induced MDDC activation may be due to the differences in receptor stimulation. LPS is known to activate toll-like receptor 4 (TLR-4) [32], however, ascophyllan receptor binding in MDDCs is not still defined. The ascophyllan receptor and associated signaling pathway will be the focus of a future study in order to better understand the ascophyllan-induced activation of MDDCs.

Unlike in vitro generated MDDCs, PBDCs comprise different subsets, which show different capacities for T cell activation [19,21]. In human PBDCs, BDCA1^+^ PBDCs are the major population and present exogenous antigen on MHC class II for the induction of CD4 T cell activation [20,21]. BDCA3^+^ PBDCs are the minor population and contribute cross-presentation for intracellular antigens on MHC class I, which induces CD8 T cell activation [19,21,33]. In the current study, we found that ascophyllan induced the activation of both BDCA1^+^ and BDCA3^+^ PBDCs. Moreover, our previous in vivo study in mice demonstrated that ascophyllan treatment promoted direct and cross-presentation of antigens in spleen DCs to CD4 and CD8 T cells. Therefore, ascophyllan may be able to promote direct and cross-presentation of antigens in both BDCA1^+^ and BDCA3^+^ PBDCs to syngeneic CD4 and CD8 T cells in humans.

Naïve CD4 and CD8 T cells differentiate into effector T cells, which produce cytokines [34]. Differentiation of IFN-γ-producing CD4 and CD8 T cells, called helper T 1 (Th1) and cytotoxic T 1 (Tc1) cells, is desirable in the development of vaccines and immunotherapies against cancer and infectious diseases [15,23]. Th1 cells further promote the activation of other immune cells, such as B cells, and induce antibody production, while Tc1 cells act as cytotoxic T lymphocytes (CTL) to kill antigen expressing cells [10,12]. Therefore, DC activation triggers an entire immune stimulation. In the current study, we demonstrated that ascophyllan promoted activation of BDCA1^+^ and BDCA3^+^ PBDCs, which secreted IL-12, the predominant cytokine for induction of Th1 and Tc1 cell differentiation. In the next study, we will evaluate the effect of ascophyllan-stimulated BDCA1^+^ and BDCA3^+^ PBDCs in the promotion of syngeneic T cell differentiation and proliferation.

Our previous study has shown that ascophyllan promoted bone marrow-derived DC and spleen DC activation in mice, in which the effect of ascophyllan was stronger than that induced by fucoidan [4]. In human study, the ascophyllan-induced activation of human DCs was not remarkably different as compared to fucoidan-induced DC activation, although the expression levels of CD86 in BDCA1^+^ PBDC and CD83 in BDCA3^+^ PBDC by ascophyllan were considerably different as compared to fucoidan treatment. These different effects of ascophyllan and fucoidan in mice and humans may be due to the receptor expression and binding affinity. Fucoidan has been shown to stimulate scavenger receptor-A (SR-A) in human DCs and the SR-A was preferentially expressed in human and mouse DCs [22,25]. However, the receptor of ascophyllan is still not clear and, therefore, it is not possible to evaluate the expression levels and affinity of the ascophyllan receptor in the receptor in human and mouse DCs. Therefore, we will analyze the ascophyllan receptor in our next study and compare the receptor expression and affinity of ascophyllan in human and mouse DCs.

## 4. Materials and Methods

### 4.1. Ethics Statement

This study was conducted according to principles expressed in the Declaration of Helsinki. Peripheral blood samples were harvested from healthy donors at the Shanghai Public Health Clinical Center. Written informed consent was obtained from all volunteers. The study has been approved by the Institutional Review Board at Shanghai Public Health Clinical Center (IRB number: 2012ZX09303013).

### 4.2. Chemicals and Cytokines

Ascophyllan was prepared from *A. nodosum* as described previously [7,8]. The ascophyllan powder was melted in phosphate buffered saline (PBS) and removed the endotoxin contamination through an endotoxin-removal filter (Zetapor Dispo, Wako, Japan). Fucoidan (*focus vesiculosus*) was purchased from Sigma-Aldrich (St. Louis, MO, USA). Recombinant human GM-CSF (rhGM-CSF) and rhIL-4 were purchased from Peprotech (Rocky Hill, NJ, USA). CSFE was obtained from Sigma-Aldrich (St. Louis, MO, USA).

### 4.3. Antibodies

Isotype control antibodies (Abs) (IgG1, IgG2a and IgG2b), Alexa647-conjugated anti-BDCA1 (IgG1, L161), PerCP/Cy5.5-conjugated anti-BDCA3 (IgG1, M80), APC/Cy7-conjugated anti-CD11c (IgG1, 3.9), Brilliant Violet510-conjugated anti-CD80 (IgG1, 2D10), PE-conjugated anti-CD83 (IgG1, HB15e), PE/Cy7-conjugated anti-CD86 (IgG2b, IT2.2), PE-conjugated HLA-DR, DP, DQ (IgG2a, Tu39) and PE/Cy7-conjugated HLA-A, B, C (IgG2a, W6/32) were obtained from BioLegend (San Diego, CA, USA). Antibodies against phosphorylated forms of p38, JNK and ERK were obtained from Cell Signaling Technology (Beverly, MA, USA).

### 4.4. Flow Cytometry

Cells were washed with PBS and incubated with unlabeled isotype control Abs and Fc-blocking antibodies for 20 min. Cells were then incubated with fluorescence-conjugated antibodies for 30 min on ice. The cells were analyzed after washing and resuspension with PBS using a FACS Aria II (Becton Dickinson, San Diego, CA, USA) and FlowJo 8.6 software (Tree Star, San Diego, CA, USA). Debris and dead cells were excluded by forward- and side-scatter gating and 4′,6-diamidino-2-phenylindole (DAPI) (Sigma-Aldrich, St. Louis, MO, USA).

### 4.5. Differentiation of MDDC and Treatment of Ascophyllan

MDDCs were generated according to an established method, with some modifications [35]. Briefly, CD14^+^ monocytes were isolated from peripheral blood mononuclear cells (PBMCs) using a CD14 positive cell isolation kit (Miltenyi Biotec, Bergisch Gladbach, Germany). Monocytes (1 × 10^6^) were incubated with 50 ng/mL rhGM-CSF and 50 ng/mL rhIL-4 in RPMI-1640 medium for 6 days, which included 10% autologous serum and 100 U/mL penicillin/streptomycin. Differentiation of immature MDDCs was defined using a flow cytometer as CD1a^+^CD14^−^ cells. Differentiated MDDCs were treated with 50 μg/mL of ascophyllan for measure co-stimulatory molecule expression and cytokine production.

### 4.6. Western Blot

The MDDCs were treated with 50 μg/mL of ascophyllan or 50 μg/mL of fucoidan for 1 h. The cell lysis buffer was prepared (20 mM pH 7.4 Tris-HCl, 50 mM NaCl, 1% Triton X-100 and protease inhibitors) and MDDCs (1 × 10^6^) were re-suspended in cell lysis buffer. Then, 10 μg of each cell lysate was subjected to sodium dodecyl sulfate-polyacrylamide gel electrophoresis (SDS-PAGE) and the proteins were transferred to a nitrocellulose membrane. Membranes were incubated with blocking buffer for 1 h and primary antibodies in blocking buffer were added for overnight at 4 °C. Secondary antibodies were applied for staining in the membranes for 1 h at room temperature (RT). Signal was detected using ECL chemiluminescence (Amersham, Uppsala, Sweden).

### 4.7. Analysis of PBDCs

PBDCs were analyzed in PBMCs using flow cytometry as previously described [21,35]. Briefly, PBMCs (10 × 10^6^) were treated with 50 μg/mL ascophyllan or 50 μg/mL fucoidan for 24 h. The PBMCs were further stained with Lineage markers, CD11c, BDCA1 and BDCA3 monoclonal antibodies in PBS. Lineage markers included FITC-conjugated CD3, CD14, CD16, CD19, CD34, CD45R, CD56 and CD235a antibodies. After incubation and washing, the cells were stained with DAPI and analyzed using flow cytometry (Becton Dickinson, San Diego, CA, USA). BDCA1^+^ and BDCA3^+^ PBDC subsets were defined from lineage^−^CD11c^+^ live leukocytes.

### 4.8. ELISA

IL-6, IL-12p70, TNF-α and IFN-γ concentrations in culture medium were measured in triplicate using ELISA kits (Biolegend, San Diego, CA, USA).

### 4.9. Statistical Analysis

Results are expressed as mean ± standard error of the mean (SEM). Data sets were analyzed by one-way ANOVA using the Tukey multiple comparison test with GraphPad Prism 4 software. *p* values less than 0.05 were considered statistically significant.

## 5. Conclusions

In conclusion, our results demonstrate that ascophyllan can promote human MDDC and PBDC activation. Thus, ascophyllan could be a promising candidate as an immune stimulatory molecule for immunotherapy in human disease such as cancer and infectious diseases.

## Figures and Tables

**Figure 1 marinedrugs-17-00066-f001:**
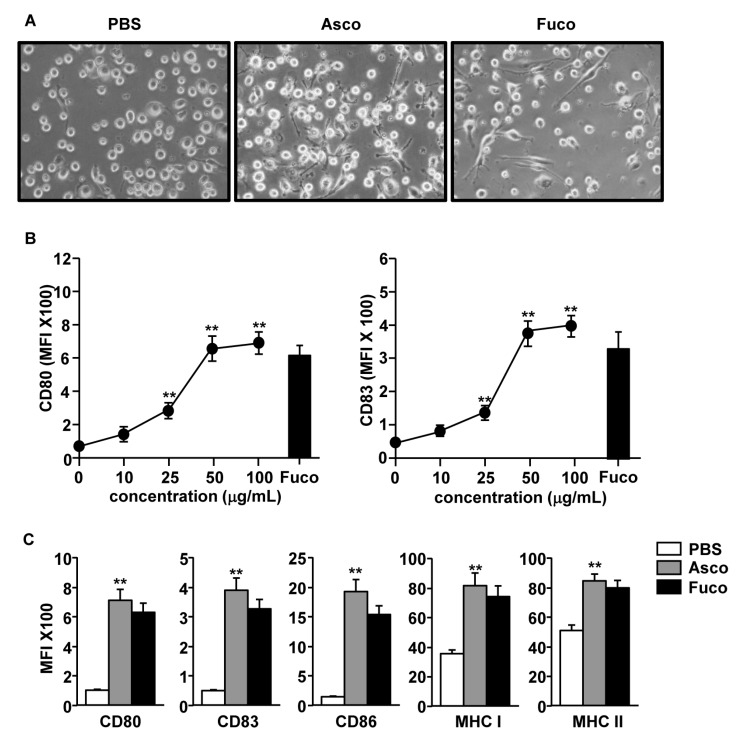
Activation of human monocyte-derived dendritic cells (MDDCs) by ascophyllan. CD14^+^ monocytes were differentiated to MDDCs by culturing with 50 ng/mL of granulocyte-macrophage colony-stimulating factor (GM-CSF) and 50 ng/mL of interleukin-4 (IL-4) for 6 days. (**A**) Changes in morphology are shown 24 h after treatment with PBS, ascophyllan (asco) or fucoidan (fuco). (**B**) Expression levels of CD80 (left panel) and CD83 (right panel) in MDDCs 24 h after treatment with different doses of ascophyllan. (**C**) Expression levels of co-stimulatory molecules were measured 24 h after treatment of PBS, ascophyllan (50 μg/mL) and fucoidan (50 μg/mL). Mean fluorescence intensity (MFI) levels of CD80, CD83, CD86, MHC class I and MHC class II in MDDCs are shown. Data are representative of or the average of analyses of 6 independent samples. ** *p* < 0.01.

**Figure 2 marinedrugs-17-00066-f002:**
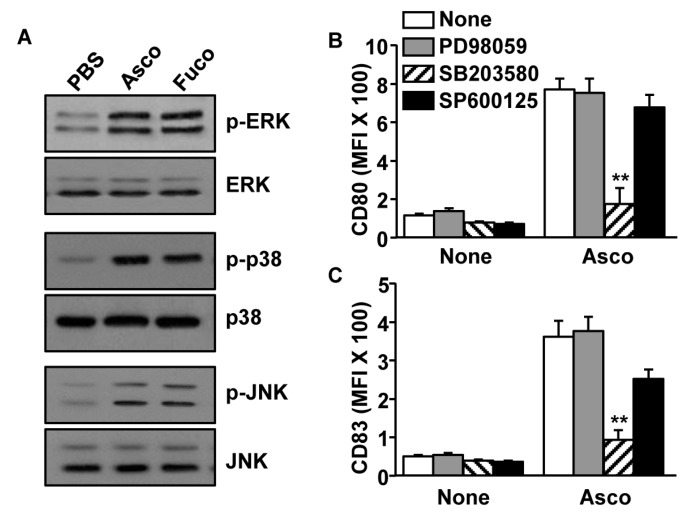
Phosphorylation of mitogen-activated protein kinase (MAPK) signaling pathway in MDDCs by ascophyllan. (**A**) Phosphorylation of ERK, p38 and JNK was analyzed in MDDCs by western blotting after 1 h of ascophyllan (50 μg/mL) or fucoidan (50 μg/mL) treatment. (B and C) MDDCs were pre-treated with PD98059 (ERK inhibitor; 10 μM), SB203580 (p38 inhibitor; 2μM) or SP600125 (JNK inhibitor; 10 μM) for 1 h and then stimulated with ascophyllan for 24 h. (**B**) MFI levels of CD80 and (**C**) CD86 are shown. All data are representative of or the average of analyses of 6 independent samples. ** *p* < 0.01.

**Figure 3 marinedrugs-17-00066-f003:**
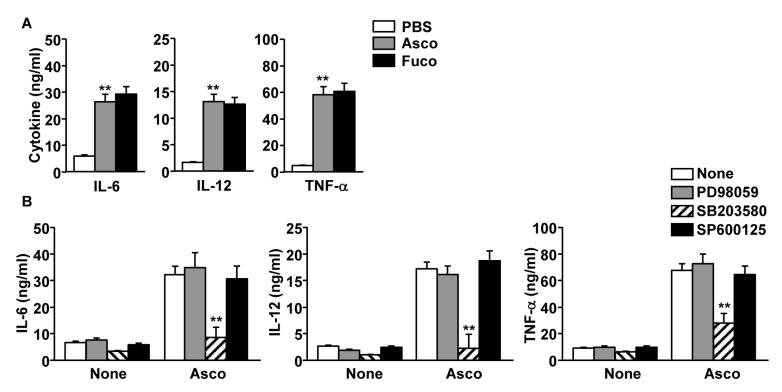
Production of pro-inflammatory cytokines from MDDCs by ascophyllan. MDDCs were treated with PBS, 50 μg/mL ascophyllan (asco) or 50 μg/mL fucoidan (fuco) for 24 h and culture medium was harvested. (**A**) The concentration of interleukin-6 (IL-6), IL-12 and TNF-α in a culture medium. (**B**) MDDCs were pre-treated with inhibitors as shown in Figure 2B,C, then cultured with PBS, ascophyllan or fucoidan for 24 h. IL-6, IL-12 and TNF-α levels were measured using ELISA. All data are the average of analyses of 6 independent samples. ** *p* < 0.01.

**Figure 4 marinedrugs-17-00066-f004:**
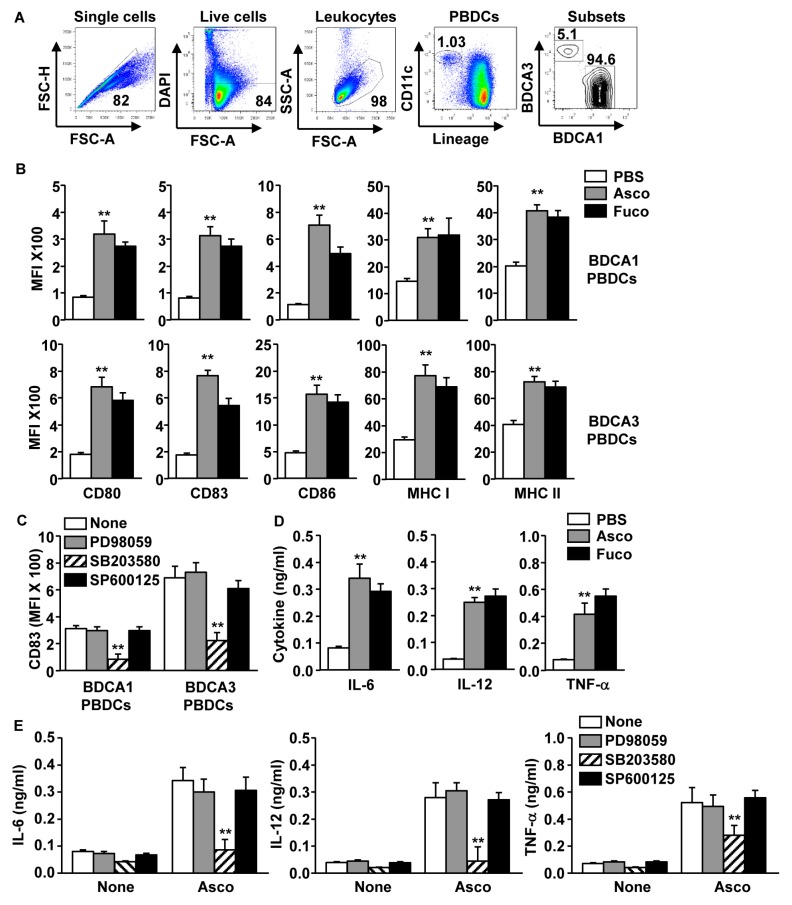
Activation of peripheral blood dendritic cell (PBDC) subsets following treatment with ascophyllan. Peripheral blood mononuclear cells (PBMCs) were cultured with PBS, 50 μg/mL ascophyllan (asco) or 50 μg/mL fucoidan (fuco) for 24 h. (**A**) BDCA1^+^ and BDCA3^+^ PBDCs were defined by flow cytometry from CD11c^+^lineage^−^ live leukocytes of PBMCs. (**B**) Co-stimulatory molecules and MHC class I and II expression in BDCA1^+^ and BDCA3^+^ PBDCs are shown. (**C**) PD98059 (ERK inhibitor; 10 μM), SB203580 (p38 inhibitor; 2 μM) or SP600125 (JNK inhibitor; 10 μM) were pre-treated in PBDCs for 1 h and the PBDCs were stimulated with 50 μg/mL ascophyllan for 24 h. Expression levels of CD83 were measured in BDCA1^+^ and BDCA3^+^ PBDCs. (**D**) Concentration of IL-6, IL-12 and TNF-α in a culture medium of PBMCs. (**E**) Concentrations of IL-6, IL-12, TNF-α in culture medium from Figure 4C were measured using ELISA. All data are representative of or the average of analyses of 6 independent samples. ** *p* < 0.01.

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
