# Peer review of "Activation of Human Dendritic Cells by Ascophyllan Purified from Ascophyllum nodosum"

_marinedrugs, 2019, doi:10.3390/md17010066_

Round 1

Reviewer 1 Report

The present paper is interesting as it describes the bioactivity of a plant compound, extracted from the algal organism Ascophyllum nodosom, on human dendritic cells. I think the paper may be published after some minor revisions:

a) fig 2 a, in Western Blotting analysis, the image of an internal loading control gene should be included.

b) why did not authors use also standard ascophyllan bought by an industrial chemical company, as further control of their own purified ascophyllan?

c) the purity degree of ascophyllan should be reported in the text.

d) in materials and methods section authors did not report any reference to which they based the techniques used in the present work. Please, address this point.

e) authors should add one or two experiments to check the toxicity level and proliferation rate of ascophyllan on their cells.

f) introduction may be enlarged by underlying the great bioactivity of plant compounds (sugars but also secondary metabolites and microRNA) on human cells… I suggest to cite the following works: Plant secondary metabolites: occurrence, structure and role in the human diet. 2008 John Wiley & Sons; Phytochemistry, 1989, 28.11: 2877-2883; PloS one2017, 12(2), e0172981.

Author Response

We thank the reviewer for the crucial comments and suggestions. We added appropriate responses for the reviewer's comments. 

The present paper is interesting as it describes the bioactivity of a plant compound, extracted from the algal organism Ascophyllum nodosom, on human dendritic cells. I think the paper may be published after some minor revisions:

Answer (A): We thank the reviewer for the comments and suggestion.

a) fig 2 a, in Western Blotting analysis, the image of an internal loading control gene should be included.

A: The control protein expression levels against phosphorylated MAPK could be total MAPK proteins. Therefore, we believe our analysis of western blotting is correct that phosphorylated MAPK levels compared with total MAPK expression levels. 

b) why did not authors use also standard ascophyllan bought by an industrial chemical company, as further control of their own purified ascophyllan?

A: As far as we know, there is currently no commercially available ascophyllan. When it comes to establishing standard for polysaccharides like ascophyllan, it is a bit difficult to find an appropriate standard, because the structures of polysaccharides differ depending on the sources, preparation procedure, and other various factors. Also, the structure deeply influences bioactivity.

c) the purity degree of ascophyllan should be reported in the text.

A: We always prepare ascophyllan from brown alga A. nodosum by the same procedure as reported previously. At the same time, we conducted composition analysis of it to make sure the quality of the ascophyllan is sufficient to get reproducible results. We believe that the purity of ascophyllan used in this study is good enough to meet the standard of naturally occurring polysaccharides. In the case of naturally occurring polysaccharides, it is almost impossible to get a completely pure sample. Even in the same algal species-derived polysaccharide, the entire structure of polysaccharide differs depending on the algal growth condition, harvest season, and the part used for preparation. In addition, polysaccharides have a certain range of molecular size distribution because they are biosynthesized products through complex processes. It is not like a protein that has a fixed structure defined by the genes.

Therefore, regarding the purity of ascophyllan, we would like to provide the composition analysis data, which is quite similar to those of previous ascophyllan.

d) in materials and methods section authors did not report any reference to which they based the techniques used in the present work. Please, address this point.

A: We have revised the Materials and Methods section.

e) authors should add one or two experiments to check the toxicity level and proliferation rate of ascophyllan on their cells.

A: In a previous study, we evaluated the toxic effect of ascophyllan in mice in vivo and found that it did not promote peripheral tissue damage or inflammation (Mar. Drugs 2014, 12(7), 4148-4164). We have also measured the toxicity of ascophyllan and fucoidan against RAW and HL-60 cells. As shown in Reviewer’s Figure 1, ascophyllan did not show any toxic effect in RAW and HL-60 cells. In contrast to ascophyllan, HL-60 cell viability was decreased by the treatment of fucoidan. It is well known that fucoidan can induce apoptosis of leukemic cells, including HL-60, NB4, and THP-1 cells. This data suggested that ascophyllan does not have a toxic effect in RAW or HL-60 cells and that the bioactivity of ascophyllan in the induction of apoptosis is different from fucoidan. We are now preparing for the next manuscript a comparison of ascophyllan and fucoidan in the induction of leukemic cell apoptosis. 

Reviewer Figure 1. Cytotoxic activity of ascophyllan in RAW and HL-60 cells. (A) RAW and (B) HL-60 cells were treated with the indicated dose of ascophyllan or fucoidan, respectively. The cell viability was measured by MTT assay.

f) introduction may be enlarged by underlying the great bioactivity of plant compounds (sugars but also secondary metabolites and microRNA) on human cells… I suggest to cite the following works: Plant secondary metabolites: occurrence, structure and role in the human diet. 2008 John Wiley & Sons; Phytochemistry, 1989, 28.11: 2877-2883; PloS one, 2017, 12(2), e0172981.

A: We thank the reviewer’s kind comments. We have now added these papers in the Introduction section.

Reviewer 2 Report

The work entitled “Activation of human dendritic cells by ascophyllan purified from Ascophyllum nodosum” describes the role of this polysaccharide on human dendritic cells by activation studies. Results evidenced that ascophyllan is able to activate dendritic cells in vitro and authors suggest that once this cells are activated, are able to activate T cells. However, the role and mechanism of DCs activating these cells is not clear. In addition, the hypothesis of this work, as described by the authors, is that “ascophyllan may also be able to induce human DC activation, and that this effect may be stronger than that of fucoidan”, however they don´t discuss about this. In most of the experiments they include fucoidan as a positive control because of its previously probed activation capacity, but they don´t compare the response induced by both polysaccharides in detail. In most cases there are no apparent differences between them, but in other cases, it seems that ascophyllan is more powerfull activating dendritic cells (i.e.: Figure 4: CD86 expression in BDCA1 PBDCs, and CD83 in BDCA3 PBDCs). Have the authors analyzed statistically if there are differences between cells treated with ascophyllan and fucoidan? Despite it does not seem that ascophyllan works much more better activating DCs than fucoidan, this analysis would contribute to support the initial hypothesis. 

The article is interesting, but there are some errors in the text and important information is missed and should be included. In addition, an important control in the MLR experiment has not been included, and this should be done in order to consider this article for publication.

In order to consider the article suitable for publication, there are several aspects to take into consideration:

- A more detailed discussion regarding the results of fucoidan in comparison with ascophyllan should be included. And if this is not the main objective of the present work, the initial hypothesis should be reformulated.

- In line 85, The capacity of MDDCs by ascophyllan…. This sentence should be reformulated. I think the word activation is missing.

- In line 102, the sentence: “We found that treatment with the ERK inhibitor, PD98058, suppressed the expression of co-stimulatory molecules that were induced by ascophyllan, while p38 and JNK inhibitors did not prevent the upregulation of these molecules in MDDCs (Figure 2B and C)”, is correct? As I see, this effect is observed with p38, not with ERK. This should be corrected. And the information regarding which of each inhibitors, inhibit each MAPK should be included.

- In results section, 2.4, each result should have the reference to its corresponding figure. 

- In the legend of the figure 3, it can be read: “Data are representative of or the average of analyses of 6 independent samples. **p < 0.01”, however I expect that data represented in this graphics is not representative of 6 experiments but the media and standard error of the mean from 6 experiments. Am I right? In this case, this figure legend should be corrected.

- There is a mistake in figure 5C legend, in case of CD4 cells, both graphics represent TNF.

- Material and methods has some errors or lack of important information:

o    The dose of both polysaccharide used in each experiment is not indicated. This is neither included in all the figure legends: figure 1 for fucoidan, figure 2 for ascophyllan, figure 3 and 4 for both polysaccharides. This information should be included in the material and methods section, and also in the figure legends.

o    The time of incubation with polysaccharides is also missed in both material and methods section, and in figure legends (figure 1 and 5). This information should be included.

o    Number of cells used for all the experiments. Authors only have included this information in sections 4.5 and 4.6. This information should be included.

o    In section 4.2 authors described that endotoxin has been removed through an endotoxin-removal filter. Has the absence of endotoxin been confirmed after this process?

o    In line 242, should it be JNK instead of AKT? 

o    In line 260, 10 mg were used for the electrophoresis. Is it right? There may be a mistake with units?

o    In line 263, in the word blocking there is a b missed.

o    In line 272, BDCA1+ and BDCA3+ cells are described to be selected as CD11c+ cells. Have been also been selected as negative for all the other lineage markers? And the same question is formulated for the next section, have been PBDCs sorted as negative for all the other lineage markers?

o    In line 275, all the antibodies were FITC-labelled?

o    In the MLR experiment, the cells (T and DCs) used for the co-culture are from the same individual? This should be clarified.

o    In the MLR experiment, the control including only T cells with ascophyllan to discard direct activation of these cells by the polysaccharide has been performed? Polysaccharides are very sticky and after washing DCs previously to the incubation with T cells, some polysaccharide can remain in the media and activate T cells unspecifically. In addition, since there is no peptidic antigen to present through MHC I nor II, no activation through antigen presentation should be expected in this case. This control is very important and must be included.

- In the discussion section (lines 203-212) and related to the previous comment, authors suggest that the activation of T cells is through MHC I or II, however, no peptides are available to present through these molecules. This discussion should be reformulated.

- In the discussion section, in line 221, authors confirm the ability of ascophyllan to differentiate naïve T cells into Th1 and Tc1 cells by activation of PBDCs, however they have used already differentiated CD4 and CD8 cells to co-culture with PBDCs (lines 282-283 in materials and methods section), so this assertion is not correct.

- In the conclusions section, authors state that the activation of MDDC and PBDC induces Th1 and Tc1 immune responses in DCs. The final part of the sentence is not clear: responses in DCs? This sentence should be reformulated. In addition, the fact that the activation of these cells induces Th1 and Tc1 immune responses should be avoided. The only effect that can been observed is that CD4 and CD8 cells are activated, and this should be confirmed performing the corresponding control.

Author Response

The work entitled “Activation of human dendritic cells by ascophyllan purified from Ascophyllum nodosum” describes the role of this polysaccharide on human dendritic cells by activation studies. Results evidenced that ascophyllan is able to activate dendritic cells in vitro and authors suggest that once this cells are activated, are able to activate T cells. However, the role and mechanism of DCs activating these cells is not clear. In addition, the hypothesis of this work, as described by the authors, is that “ascophyllan may also be able to induce human DC activation, and that this effect may be stronger than that of fucoidan”, however they don´t discuss about this. In most of the experiments they include fucoidan as a positive control because of its previously probed activation capacity, but they don´t compare the response induced by both polysaccharides in detail. In most cases there are no apparent differences between them, but in other cases, it seems that ascophyllan is more powerfull activating dendritic cells (i.e.: Figure 4: CD86 expression in BDCA1 PBDCs, and CD83 in BDCA3 PBDCs). Have the authors analyzed statistically if there are differences between cells treated with ascophyllan and fucoidan? Despite it does not seem that ascophyllan works much more better activating DCs than fucoidan, this analysis would contribute to support the initial hypothesis. 

The article is interesting, but there are some errors in the text and important information is missed and should be included. In addition, an important control in the MLR experiment has not been included, and this should be done in order to consider this article for publication.

Answer (A): We thank the reviewer for the comments and suggestion. As the reviewer has mentioned, we revised the Discussion and Materials and Methods sections.  

In order to consider the article suitable for publication, there are several aspects to take into consideration:

- A more detailed discussion regarding the results of fucoidan in comparison with ascophyllan should be included. And if this is not the main objective of the present work, the initial hypothesis should be reformulated.

A: We thank the reviewer for pointing this out. We have now revised the description of the discussion section as follows: “Our previous study has shown that ascophyllan promoted bone marrow-derived DC and spleen DC activation in mice, in which the effect of ascophyllan was stronger than that induced by fucoidan {Zhang, 2014 #50}. In human study, the ascophyllan-induced activation of human DCs was not remarkably different as compared to fucoidan-induced DC activation, although the expression levels of CD86 in BDCA1+ PBDC and CD83 in BDCA3+ PBDC by ascophyllan were considerably different as compared to fucoidan treatment. These different effects of ascophyllan and fucoidan in mice and humans may be due to the receptor expression and binding affinity. Fucoidan has been shown to stimulate scavenger receptor-A (SR-A) in human DCs and the SR-A was preferentially expressed in human and mouse DCs {Jin, 2009 #20; Jin, 2014 #21}. However, the receptor of ascophyllan is still not clear and therefore it is not possible to evaluate the expression levels and affinity of the ascophyllan receptor in human and mouse DCs. Therefore, we will analyze the ascophyllan receptor in our next study and compare the receptor expression and affinity of ascophyllan in human and mouse DCs”

- In line 85, The capacity of MDDCs by ascophyllan…. This sentence should be reformulated. I think the word activation is missing.

A: We have now reformulated.

- In line 102, the sentence: “We found that treatment with the ERK inhibitor, PD98058, suppressed the expression of co-stimulatory molecules that were induced by ascophyllan, while p38 and JNK inhibitors did not prevent the upregulation of these molecules in MDDCs (Figure 2B and C)”, is correct? As I see, this effect is observed with p38, not with ERK. This should be corrected. And the information regarding which of each inhibitors, inhibit each MAPK should be included.

A: We are sorry for the error. It is p38 and its inhibitor, SB203580. We also added information for each inhibitor in the manuscript.

- In results section, 2.4, each result should have the reference to its corresponding figure. 

A: We have now added the references.

- In the legend of the figure 3, it can be read: “Data are representative of or the average of analyses of 6 independent samples. **p < 0.01”, however I expect that data represented in this graphics is not representative of 6 experiments but the media and standard error of the mean from 6 experiments. Am I right? In this case, this figure legend should be corrected.

A: We have now revised the figure legend.

- There is a mistake in figure 5C legend, in case of CD4 cells, both graphics represent TNF.

A: We have now corrected this.

- Material and methods has some errors or lack of important information:

o    The dose of both polysaccharide used in each experiment is not indicated. This is neither included in all the figure legends: figure 1 for fucoidan, figure 2 for ascophyllan, figure 3 and 4 for both polysaccharides. This information should be included in the material and methods section, and also in the figure legends.

A: We have now added the information in materials and methods and figure legends.

o    The time of incubation with polysaccharides is also missed in both material and methods section, and in figure legends (figure 1 and 5). This information should be included.

A: We added incubation time in the Materials and Methods section and in the Figure legends.

o    Number of cells used for all the experiments. Authors only have included this information in sections 4.5 and 4.6. This information should be included.

A: We have now added cell numbers in Materials and Methods section.

o    In section 4.2 authors described that endotoxin has been removed through an endotoxin-removal filter. Has the absence of endotoxin been confirmed after this process?

A: Yes, we have confirmed the endotoxin levels after removal to be below 0.1 EU/μg.

o    In line 242, should it be JNK instead of AKT? 

A: We have now corrected this.

o    In line 260, 10 mg were used for the electrophoresis. Is it right? There may be a mistake with units?

A: We have now corrected this. It is 10 μg.

o    In line 263, in the word blocking there is a b missed.

A: We have now corrected this.

o    In line 272, BDCA1+ and BDCA3+ cells are described to be selected as CD11c+ cells. Have been also been selected as negative for all the other lineage markers? And the same question is formulated for the next section, have been PBDCs sorted as negative for all the other lineage markers?

A: Yes, we gated Lineage-CD11c+ cells for analysis of PBDC activation and sorting of the PBDCs.

o    In line 275, all the antibodies were FITC-labelled?

A: Those antibodies were all conjugated with FITC. The FITC-expressing cells defined as lineage-positive cells, which are not DCs.

o    In the MLR experiment, the cells (T and DCs) used for the co-culture are from the same individual? This should be clarified.

A: No, T cells and DCs were not from the same individual.

o    In the MLR experiment, the control including only T cells with ascophyllan to discard direct activation of these cells by the polysaccharide has been performed? Polysaccharides are very sticky and after washing DCs previously to the incubation with T cells, some polysaccharide can remain in the media and activate T cells unspecifically. In addition, since there is no peptidic antigen to present through MHC I nor II, no activation through antigen presentation should be expected in this case. This control is very important and must be included.

A: For evaluation of the capacity of ascophyllan-stimulated DCs in T cell activation, the PBDCs were treated with ascophyllan for 24 hours. The stimulated PBDCs were then washed to remove free ascophyllan in the medium. In addition, we examined the ascophyllan effect on isolated T and NK cells and found the ascophyllan induced NK cell activation, but not T cell activation. Since ascophyllan cannot induce T cell activation, the ascophyllan-stimulated PBDC induced proliferation of T cells is not mediated by any remaining ascophyllan in the medium. We are preparing another manuscript that compares ascophyllan and fucoidan in NK and T cell activation. Therefore, we added the results only for reviewers (Reviewer’s Figure 2).

Reviewers Figure 2. Ascophyllan-induced production of IFN-γ in natural killer (NK cells). (A) Isolated NK cells from peripheral blood were treated with PBS or 50 μg/ml ascophyllan for 6 hours. Intracellular cytokine production was measured in NK cells. (B and C) CD4 and CD8 T cells were isolated from peripheral blood and cultured with PBS or ascophyllan in the presence or absence of anti-CD3 and anti-CD28 coatings in plate. Three days after the treatment of ascophyllan, intracellular IFN-γ and IL-17 levels were measured in CD4 and CD8 T cells, respectively.  

As the reviewer mentioned, antigen presentation is the most important function in DC activation and our data cannot fully support ascophyllan as promoting an antigen-specific immune response. However, in human study, antigen peptide-specific T cell proliferation and activation assay is not easy to perform compared to mouse study. There are many experimental tools for the measurement of antigen-specific immune response in mice, such as OVA specific mice and OVA tetramer, and we also showed ascophyllan promoted antigen-specific immune responses in mice in vivo (Oncotarget. 2016 Apr 12;7(15):19284-98). However, in human study, specific antigen mutated patients (HLA-A2+ donors) are required to measure antigen-specific immune responses. Although we did not show antigen-specific T cell proliferation, ascophyllan-stimulated PBDCs promoted T cell proliferation and IFN-γ production. Therefore, ascophyllan may be a candidate of adjuvant for enhancing vaccine activities. 

- In the discussion section (lines 203-212) and related to the previous comment, authors suggest that the activation of T cells is through MHC I or II, however, no peptides are available to present through these molecules. This discussion should be reformulated.

A: As we presented MHC I and II expression data, the ascophyllan promoted up-regulation of those molecules. Although these data cannot prove the antigen-specific T cell activation, the up-regulated MHC class I and II and co-stimulatory molecules on DCs were enough to promote the proliferation and activation of T cells.

- In the discussion section, in line 221, authors confirm the ability of ascophyllan to differentiate naïve T cells into Th1 and Tc1 cells by activation of PBDCs, however they have used already differentiated CD4 and CD8 cells to co-culture with PBDCs (lines 282-283 in materials and methods section), so this assertion is not correct.

A: Naïve T cells are non-stimulated T cells, which do not produce cytokines. To clarify, we revised the sentence. For T cell stimulation assay, we isolated non-stimulated, fresh T cells from a healthy donor.

- In the conclusions section, authors state that the activation of MDDC and PBDC induces Th1 and Tc1 immune responses in DCs. The final part of the sentence is not clear: responses in DCs? This sentence should be reformulated. In addition, the fact that the activation of these cells induces Th1 and Tc1 immune responses should be avoided. The only effect that can been observed is that CD4 and CD8 cells are activated, and this should be confirmed performing the corresponding control.

A: As we mentioned above, ascophyllan did not induce T cell proliferation and activation. Therefore, T cell activation in this study is mediated by ascophyllan-stimulated DCs. 

Round 2

Reviewer 2 Report

The revised version of the manuscript has been improved, however, there are still two questions that concern me:

- I understand that DC have been selected as CD11c+, but additional lineage markers from cells that are not DCs have been included (CD3, CD14, CD16, etc., ). I would like to know if the selection for DCs sorting has been performed as CD11c+ but also as negative for all the other lineage markers described.

- Regarding the MLR experiment, since the co-culture is allogenic, the activation could have been indirect and not specific; in this case mediated by an allogenic response rather than because of the specific polysaccharide by itself. This is supported by the fact that focoidan has more or less the same effect as ascophyllan. In this sense, ascophyllan could have increased the expression of MHCII molecules (in fact, this was described by the authors) and this could have triggered an allogenic response that would not be observed if these DCs would be co-cultured with cells from the same individual, or an individual with more similar MHCII molecule haplotype. This should be clarified in the text, or the experiment removed from the manuscript.

Author Response

Reviewer 2

The revised version of the manuscript has been improved, however, there are still two questions that concern me:

- I understand that DC have been selected as CD11c+, but additional lineage markers from cells that are not DCs have been included (CD3, CD14, CD16, etc., ). I would like to know if the selection for DCs sorting has been performed as CD11c+ but also as negative for all the other lineage markers described.

Answer (A): We thank the reviewer for the comments and suggestion. Since the study was focused on PBDCs, we have not isolated CD11c-Lineage- cell population, which may be different immune cells. Moreover, we have now deleted PBDC sorting data as the reviewer was recommended in next question. Therefore, we will compare CD11c+lineage- and CD11c-lineage- cells in syngeneic T cell activation in our next study.

- Regarding the MLR experiment, since the co-culture is allogenic, the activation could have been indirect and not specific; in this case mediated by an allogenic response rather than because of the specific polysaccharide by itself. This is supported by the fact that focoidan has more or less the same effect as ascophyllan. In this sense, ascophyllan could have increased the expression of MHCII molecules (in fact, this was described by the authors) and this could have triggered an allogenic response that would not be observed if these DCs would be co-cultured with cells from the same individual, or an individual with more similar MHCII molecule haplotype. This should be clarified in the text, or the experiment removed from the manuscript.

A: We thank the reviewer’s kind comments. We have carefully checked other paper and clearly understood what the reviewer’s ask us to revise. We agreed the reviewer’s suggestion and decide to delete allogenic T cell activation data. In our next study, we will evaluate the ascophyllan effect in syngeneic T cell activation by stimulation of PBDCs.

Round 3

Reviewer 2 Report

I think now the manuscript is suitable for publication, however authors should revise the following mistakes:

- In lines 204-205, the following sentence is not complete: "Therefore, we will evaluate the differentiation of syngeneic T cells by ascophyllan- treated PBDCs in our next..."

- In lines 268-270, if the authors have not used the data provided by the non-DCs lineage markers, the following sentence, should be removed: "Lineage staining included FITC-conjugated CD3, CD14, CD16, CD19, CD34, CD45R, CD56, and CD235a antibodies"

- The section "4.8 Isolation of PBDCs" in material and methods has not been removed.

- In conclusions section: "In conclusion, our results demonstrate that ascophyllan can promote human MDDC and PBDC activation, which the activated DCs induced T cell proliferation and IFN-γ production", this last sentence should also be removed.

As the authors have deleted allogenic T cell activation data, there is a need to carefully revise the manuscrip to confirm that there are not remaining comments regarding these data.

Author Response

Reviewer 2

I think now the manuscript is suitable for publication, however authors should revise the following mistakes:

- In lines 204-205, the following sentence is not complete: "Therefore, we will evaluate the differentiation of syngeneic T cells by ascophyllan- treated PBDCs in our next..."

Answer (A): Thank you very much for your thorough review. We really appreciated it. We have now revised the sentence.

- In lines 268-270, if the authors have not used the data provided by the non-DCs lineage markers, the following sentence, should be removed: "Lineage staining included FITC-conjugated CD3, CD14, CD16, CD19, CD34, CD45R, CD56, and CD235a antibodies"

A: For analysis of PBDC subsets, we gated single cells, live cells and leukocytes. Next, we gated Lineage-negative and CD11c-positve cell population. The lineage markers were included CD3, CD14, CD16, CD19, CD34, CD45R, CD56 and CD235a. These marker-expressing cells are not DCs, therefore we gated lineage-negative and CD11c-positive cells as the PBDCs. Moreover, we further divided the PBDCs to BDCA1-postive and BDCA3-positive cells. You can fine flow cytometry strategy for PBDC analysis in Figure 4A. Therefore, we believe that we have to mention what kind of markers were used for lineage staining in the Materials and Methods.

- The section "4.8 Isolation of PBDCs" in material and methods has not been removed.

A: We are sorry for the error. We have now deleted it.

- In conclusions section: "In conclusion, our results demonstrate that ascophyllan can promote human MDDC and PBDC activation, which the activated DCs induced T cell proliferation and IFN-γ production", this last sentence should also be removed.

A: We revised the sentence.

As the authors have deleted allogenic T cell activation data, there is a need to carefully revise the manuscrip to confirm that there are not remaining comments regarding these data.

A: We have carefully revised the manuscript. We thanks for your kind comments.
